# Metabolic Characterization of Two Flor Yeasts During Second Fermentation in the Bottle for Sparkling Wine Production

**DOI:** 10.3390/ijms262110457

**Published:** 2025-10-28

**Authors:** Juan Carlos García-García, María Trinidad Alcalá-Jiménez, Juan Carlos Mauricio, Cristina Campos-Vázquez, Inés M. Santos-Dueñas, Juan Moreno, Teresa García-Martínez

**Affiliations:** 1Department of Agricultural Chemistry, Edaphology and Microbiology, Microbiology Area, Severo Ochoa Building (C6), Campus of Rabanales, Agrifood Campus of International Excellence ceiA3, Universidad de Córdoba, Ctra. N-IV-A, Km 396, 14014 Córdoba, Spain; p22gagaj@uco.es (J.C.G.-G.); b52aljim@uco.es (M.T.A.-J.); qe1movij@uco.es (J.M.); mi2gamam@uco.es (T.G.-M.); 2Department of Inorganic Chemistry and Chemical Engineering, Chemical Engineering Area, Marie Curie Building (C3), Campus of Rabanales, Chemical Institute for Energy and Environment (IQUEMA), Agrifood Campus of International Excellence ceiA3, Universidad de Córdoba, Ctra. N-IV-A, Km 396, 14014 Córdoba, Spain; q42cavac@uco.es (C.C.-V.); ines.santos@uco.es (I.M.S.-D.)

**Keywords:** sparkling wine, second fermentation, flor yeast, volatile compounds, nitrogen compounds

## Abstract

The global sparkling wine market continues to grow steadily, reaching approximately 24 million hectoliters in 2023, with an annual increase of around 4% despite a general decline in overall alcoholic beverage consumption. This growth highlights the importance of employing diverse yeast strains to improve product variety and quality. Flor yeasts are specialized strains of *Saccharomyces cerevisiae* that develop a biofilm on the surface of certain wines during biological ageing. They possess unique physiological properties, including high ethanol tolerance and the capacity to adhere, which supports wine clarification. They also have the ability to contribute unique volatile compounds and aroma profiles, making them promising candidates for sparkling wine production. This study evaluated two flor yeast strains (G1 and N62), which were isolated from the Pérez Barquero winery during the second fermentation process using the traditional method. Sparkling wines were produced by inoculating base wine (BW) with each strain, and the wines were monitored at 3 bar CO_2_ pressure and after 9 months of ageing on lees. Comprehensive metabolomic analysis was performed using GC-MS for volatile compounds and HPLC for nitrogen compounds, with statistical analysis including PCA, ANOVA, Fisher’s LSD, and correction FDR tests. Strain N62 demonstrated faster fermentation kinetics and higher cellular concentration, reaching 3 bar pressure in 27 days compared to 52 days for strain G1. Both strains achieved similar final pressures, 5.1–5.4 bars. Metabolomic profiling revealed significant differences in the profiles of volatile and nitrogen compounds between the two strains. G1 produced higher concentrations of 3-methyl-1-butanol, 2-methyl-1-butanol, and acetaldehyde, while N62 generated elevated levels of glycerol, ethyl esters, and amino acids, including glutamic acid, aspartic acid, and alanine. These findings demonstrate that both flor yeast strains successfully complete sparkling wine fermentation while producing distinct metabolic signatures that could contribute to unique sensory characteristics. This supports their potential as alternatives to conventional sparkling wine yeasts for enhanced product diversification.

## 1. Introduction

Over the past two decades, the global wine market has undergone notable transformations. Although total wine consumption continues to decline, the demand for sparkling wines has expanded sharply, drawing increasing attention from producers and analysts. According to the latest report from the International Organization of Vine and Wine (OIV, [1]), global sparkling wine production reached approximately 24 million hL in 2023, representing around 8.5% of total world wine output and showing an average annual growth of 4% over the last decade. Overall global demand for sparkling wines has risen by about 57% in recent years, with Asia accounting for one of the fastest-growing markets, driven by higher disposable income and evolving consumer preferences. The global market was valued at USD 47.57 billion in 2024 and is projected to reach USD 69.22 billion by 2032 at a compound annual growth rate (CAGR) of 4.8%. This sustained expansion is largely explained by changing consumption habits favoring lighter, more refreshing alcoholic beverages with moderate alcohol content and vibrant acidity. Sparkling wines are increasingly consumed not only during celebrations but also as casual aperitifs or as bases for modern cocktails, reinforcing their everyday appeal. These shifts are supported by broader product availability, more accessible price ranges, and an overarching consumer preference for higher quality products [2].

A persistent challenge in the production of sparkling wines is the restricted availability of commercial yeast strains for in-bottle secondary fermentation, which limits product diversity. Using native (autochthonous) yeast strains ensures that wines express regional typicity while contributing to differentiation in global markets. Vigentini et al. [3] highlighted the significance of characterizing and selecting indigenous yeasts as a strategic tool for diversification. In this context, flor yeasts—specialized *Saccharomyces cerevisiae* strains capable of forming a buoyant biofilm (the flor veil) during biological ageing—represent promising candidates for innovative sparkling wine applications. Their distinctive physiological and metabolic traits, including high ethanol tolerance, strong surface adhesion that favors clarification, and the production of complex volatile profiles, make them particularly valuable to the industry [4,5].

In this study, the traditional Méthode Champenoise was employed, in which dedicated yeast strains ferment the base wine (BW) within sealed bottles, producing endogenous CO_2_ pressure [6,7,8]. Following this second fermentation, the wine is aged on lees for periods ranging from 9 months to several years [9,10]. Selecting appropriate yeast strains for this process is therefore a decisive factor in shaping the sensory profile of sparkling wines [3,11]. The targeted use and characterization of autochthonous yeasts have gained commercial importance for enhancing each terroir’s uniqueness as these microorganisms adapt to regional environmental conditions, contributing to metabolic and organoleptic differentiation [12,13,14].

From an oenological perspective, desirable properties such as reduced ethanol levels, enriched glycerol content, and the formation of key metabolites like diethyl succinate—which serves as an ageing marker—depend on yeast strain selection [15,16,17]. Moreover, lees ageing protects wines from oxidation and browning, while enzymatic strategies (e.g., β glucanase applications) offer accelerated maturation and improved antioxidant potential [6,18,19]. Nitrogen compounds released during autolysis play a crucial role in the chemical and sensory evolution of sparkling wines [20,21,22], acting as precursors for higher alcohols and esters via amino acid metabolism and Maillard-type reactions. These transformations generate the toasted, nutty, and caramel nuances often associated with complexity in mature sparkling wines [23,24,25,26].

The aim of this research is to introduce unconventional flor yeast strains for sparkling wine production, capable of producing distinct metabolic signatures while maintaining quality standards typical of premium products. This work represents the first integrated metabolomic and nitrogen compound profiling of flor yeasts during in-bottle second fermentation, extending beyond earlier proteomic and volatilomic studies. The findings provide a comprehensive metabolic characterization of these flor yeasts, establishing their strain-specific attributes and potential for future industrial application.

## 2. Results

### 2.1. CO_2_ Pressure Monitoring of Second Fermentation and General Oenological Parameters

Two flor yeast strains of *S. cerevisiae* (G1 and N62) were isolated from flor veils in the Pérez Barquero winery. They were characterized as described in the ”Materials and Methods” section and evaluated for their fermentative performance. Figure 1 shows how CO_2_ pressure produced by each yeast strain changed during the second fermentation in the bottle. Strain G1 achieved 3 bars of pressure after 52 days, reaching 5.1 bars. The N62 strain achieved 3 bars of pressure after 27 days, reaching 5.4 bars. These results demonstrate that the N62 strain exhibited a higher fermentation rate than G1.

Table 1 shows the general oenological parameters of the strains at both sampling points. Table 2 presents the yeast cell counts (viable and total cells) of the wines obtained.

The main differences observed between the two strains were that N62 achieved a higher cell count and cell viability. This was reflected in higher volatile acidity; this parameter showed two HGs differentiating between the two strains and lower concentrations of malic and lactic acids. This suggests that N62 may be better adapted to, and more resistant to, the harsh conditions of overpressure.

### 2.2. Changes in the Metabolic Profiles of Different Wines

Two principal component analyses (PCAs) were performed on a data matrix consisting of 77 metabolites quantified as variables (Appendix A). According to the results shown in Figure 2a,b, there is a great differentiation between the replicates of the different wines studied. The PC1 and PC2 axes explain 88% of the total variance for Figure 2a, and 90.5% of the total variance is explained by the axes, according to Figure 2b.

To better visualize changes in metabolites across different wines for each strain, several heat maps were generated using normalized data, as illustrated in Figure 3. As can be seen from the heat maps, all triplicates are more similar to each other than to the other samples. Regarding the wine made with strain G1 (Figure 3a–c), peaks of acetoin quantification were detected at 3 bars of pressure, and peaks of ethyl lactate for the BW and a higher number of quantification peaks were observed for the rest of the major volatile compounds at 9 months, as shown in Figure 3a. In Figure 3b, a high quantification of the minor volatile compounds ethyl octanoate, 2-furanmethanol, 4-ethylguaiacol, acetophenone, ethyl hexadecanoate, pentylfuran, ethyl tetradecanoate, and ethyl dodecanoate was detected for the BW; furfural, limonene, hexanol, decanal, nonanal, 2-phenylethanol acetate, ethyl butanoate, ethyl decanoate, and cis-3-hexenyl butyrate for the wine with G1 at 3 bar pressure; and the rest of the compounds for the wine at 9 months. Concerning non-volatile metabolites, the nitrogen compounds quantified in the wines inoculated with strain G1 evolved as shown in Figure 3c. Putrescine, L-leucine, L-glutamic acid, L-lysine, L-aspartic acid, L-ornithine, and L-valine were quantified in higher proportions in the BW; the remaining non-volatile compounds were detected in wine with G1 at 9 months. However, the lowest quantifications of all nitrogen compounds were found in the G1 wine at 3 bar pressure. In the wine with strain N62 (Figure 3d–f), peaks of acetoin, acetaldehyde, 1,1-diethoxyethane, 2,3-butanediol (meso and levo), 3-metil-1-butanol, and methanol quantification were detected for the wine with N62 at 3 bars of pressure, and regarding the rest of the major volatile compounds, a higher quantification was observed for the wine with N62 at 9 months, as shown in Figure 3d. In Figure 3e, a high quantification of the minor volatile compounds acetophenone, Z-3-hexenol acetate, ethyl hexadecanoate, acetate hexyl, and octanol was observed for the BW; ethyl octanoate, limonene, 2-phenylethanol acid, decanal, and nonanal for the wine with N62 at 3 bar pressure; and the rest of the compounds for the wine with N62 at 9 months. For the non-volatile metabolites, amino acids, biogenic amines, and ammonium ions quantified in the wines inoculated with the highest proportion were L-lysine and L-proline in the BW; wine inoculated with N62 at 3 bar pressure was found to have high L-valine, ammonium chloride, L-glutamic acid, putrescine, and tyramine quantifications; and the rest of the nitrogen compounds quantified in the wine with N62 at 9 months evolved as shown in Figure 3f.

### 2.3. Metabolic Analysis Throughout the Second Fermentation

Regarding the major volatile compounds, only quantitative differences were observed; however, among the minor compounds both qualitative and quantitative differences were found. Notably, acetophenone was detected exclusively in the BW, whereas geranyl acetate was identified only in the wine fermented with strain N62. (The results are provided in the Appendix A). In the nitrogen compounds, quantitative and qualitative differences were found; phenylalanine was only quantified in wine with strain G1 and glycine, and agmatine sulphate salt and L-norleucine were only identified in wine with strain N62. (The results are provided in the Appendix A).

#### 2.3.1. For Wines Inoculated with Strain G1

Figure 4a shows a one-way analysis of variance (ANOVA), a Fisher’s LSD test, and correction FDR test < 0.05 for the major volatile compounds between the BW and the wines inoculated with strain G1. Significant differences were observed for all compounds except for ethyl acetate and 2,3-butanediol (levo). For ethyl lactate, no differences were found between the wines with G1; moreover, only acetoin was significantly more quantified in the wine with G1 at 3 bar pressure than in the wine at 9 months (see Figure 4d). All other compounds were found in higher concentration in the wine with G1 at 9 months. A total of 42 minor volatile compounds were identified in Figure 4b, of which 36 showed significant differences between the BW and the wines inoculated with strain G1. These compounds are 2 furanoics, 13 esters, 7 alcohols, 2 ketones, 4 aldehydes, 2 terpenoides, 5 acetate, and 1 lactone. A total of 20 compounds were upregulated for wine with G1 at 9 months with respect to wine at 3 bar pressure, whereas 9 compounds were downregulated. In addition, five compounds were upregulated for the BW with respect to the other two wines (see Figure 4e). A total of 15 nitrogen compounds were identified with significant differences, as shown in Figure 4c. Twelve of these compounds were found to be upregulated for the G1 wine at 9 months relative to the wine at 3 bar pressure, and the other three compounds were upregulated for the BW relative to the G1 wine (see Figure 4f).

#### 2.3.2. For Wines Inoculated with Strain N62

Figure 5a compares the major volatile compounds between the BW and the wines inoculated with the strain N62. Significant differences were observed in all compounds. The significant compounds with the highest concentration for the wine with N62 at 3 bar pressure were 3-methyl-1-butanol, methanol, acetaldehyde, acetoin, 1,1-diethowyethane, and 2,3-butanediol (meso and levo); the remaining compounds were found at significantly higher concentrations for the wine at 9 months (see Figure 5d). A total of 43 minor volatile compounds were identified in Figure 5b, of which 40 presented significant differences between the BW and the wines inoculated with strain N62. These compounds are 17 esters, 7 alcohols, 5 aldehydes, 2 ketones, 5 acetates, 2 terpenoides, 1 lactone, and 1 furanoic. A total of 29 compounds were upregulated for wine with N62 at 9 months with respect to wine at 3 bar pressure, whereas 5 compounds were downregulated. Furthermore, two compounds were upregulated for the BW with respect to the other two wines (see Figure 5e). A total of 16 nitrogen compounds were quantified with significant differences, as shown in Figure 5c, and L-norleucine and glycine only appear in the N62 wines, both compounds being significantly higher for the wine at 9 months. Five compounds were found to be upregulated for the N62 wine at 9 months compared to the wine at 3 bar pressure, and three other compounds were upregulated for the wine at 3 bar pressure. Finally, only one compound was found to be significantly higher for the BW than for the other two wines (see Figure 5f).

### 2.4. Comparison of the Metabolomic Profile of Wines at 9 Months

Metabolites identified in 9-month wines from both strains were compared using one-way ANOVA and Tukey’s HSD tests (q < 0.05). After statistical filtering, nine major volatile compounds displayed significant differences: five were upregulated in wines produced with strain G1, while four were elevated in those made with strain N62 (Figure 6a). In Figure 6b, statistical analysis revealed 13 minor volatile compounds without significant differences, in addition to 7 compounds upregulated in G1 wines and 18 in N62 wines. Notably, geranyl acetate, pentylfuran, and 4-ethylguaiacol were exclusively identified in strain N62 wines, whereas z-3-hexanol acetate was unique to G1 wines. Figure 6c shows that, among nitrogen-containing compounds, 2 exhibited no significant differences, 10 were upregulated for N62, and only 1 for G1. L-valine was absent in both strains’ wines at 9 months. Furthermore, glycine, agmatine sulphate salt, and L-norleucine were detected only in wines fermented with strain N62, while L-phenylalanine was identified exclusively in G1 wines (Figure 6c).

## 3. Discussion

Sparkling wine represents a complex, high-value alcoholic beverage produced through the second fermentation of a base still wine, followed by an extended maturation phase that can vary from a minimum of 9 months to several years, depending on the production method and desired sensory profile. Many of the components responsible for the metabolic and organoleptic characteristics of wine are derived from the alcoholic fermentation of yeast and are influenced by the type of strain used [11,27].

Alcohols are biosynthesized during fermentation from sugar metabolism or amino acid catabolism via the Ehrlich pathway [28]. 3-Methyl-1-butanol may be produced from leucine during fermentation by its deamination and decarboxylation [29]. This major volatile compound showed significant differences throughout the fermentation in both strains, but the wine with the G1 strain produced a significantly higher concentration. It seems that G1 has a greater capacity to synthesize this volatile compound. 2-Methyl-1-butanol, together with the aforementioned alcohol, contributes fruity aromas to wines [30] and was also formed in significantly higher proportions in wines produced by strain G1 than in the other strains. 2-Phenylethanol can be produced through the transamination of L-phenylalanine via the Ehrlich pathway. This process may contribute to the rose aroma of wine [31]. No significant differences were found at the end of fermentation between the two wines with either strain, although their concentration did increase over time (see Figure 4b and Figure 5b). The amino acid L-phenylalanine was only found in the G1 wine at 9 months, so it appears that this strain has the potential to produce more of this main volatile. 2-Methyl-1-propanol (or isobutanol) can be synthesized by the degradation of valine and may contribute herbaceous notes to wines [32]. This main volatile compound increased significantly in wines from both strains throughout fermentation, but at the end of this process, no significant differences were found between the two wines. At the same time of fermentation, both strains completely consumed the valine. 1-Hexenol is a minor volatile compound, specifically a higher alcohol that may give wine a vegetal or herbaceous aroma; in high quantities it is not a pleasant smell [33], but the perception threshold was not exceeded in any of the wines, and in the wine with N62 it increased significantly compared to the wine with G1, which is an expected result according to Jagatić Korenika et al. [34]. This compound can be transformed into a hexyl acetate, by yeast metabolism [4]. In wine with G1 it increases significantly throughout fermentation, whereas in wine with N62 it decreases, so it seems that this strain did not carry out this transformation under these conditions. Methanol is a major volatile compound; it is not a fermentation byproduct. It originates from grape pectin, the methoxyl groups of which are hydrolyzed by endogenous pectinases. It can be toxic at high concentrations, but none of the wines studied exceeded the perception threshold [35]. Of the carbonyl compounds that make up the major volatile compounds, only acetaldehyde showed significant positive differences in wine with G1 at 9 months. However, this compound was found in both wines within the perception threshold, which may contribute to a fruity aroma without being unpleasant [36]. This carbonyl compound can be formed from alanine [37]. This amino acid was found in significantly high quantities in wine at 9 months with strain N62 than with strain G1, suggesting that strain N62 may have a lower conversion capacity. Acetaldehyde is also a metabolite that can be transformed into acetate, acetoin, and ethanol [38]. As the results show, acetaldehyde decreased significantly in both wines at 9 months, and ethanol increased in both wines but more significantly in the wine with the G1 strain. Furthermore, 1,1-diethoxyethane can be formed from acetaldehyde and ethanol [39], which also increased significantly in the G1 wine at 9 months. There are carbonyl compounds that are part of the minor volatile compounds. One aldehyde is hexanal; this metabolite was found before the start of fermentation and at 9 months but not during, and it is possible that it was being transformed into higher alcohols, as occurred in Ubeda et al. [16]. At 9 months, this compound showed no significant differences between the wines, but it did exceed the threshold of perception and thus may contribute aroma to the wine. Octanal was quantified as upregulated for wine with N62 at 9 months but did not exceed the perception threshold. Decanal aldehyde did not show significant differences between wines at 9 months but exceeded the odor threshold, and nonanal did show differences, being found in greater proportion in wine with N62, thus exceeding the perception threshold; these data agree with those presented by de Lerma et al. [40], which report that these compounds provide a citrus aroma. Most ketones and aldehydes that are part of the minor volatile compounds increase their concentration significantly throughout fermentation, as occurred in the investigations of Jagatić Korenika et al. [34]. At 9 months all these metabolites, except two that did not present significant differences, were found to be upregulated for the N62 wine.

Polyols are semi-volatile or non-volatile compounds that depend on fermentable sugars, products derived from yeast metabolism and other factors [41]. Glycerol contributes to the smoothness and viscosity of wine [42]. This compound intervenes to improve redox balance and hyperosmotic stress in yeasts [43]. As can be seen in Figure 6a, the N62 strain seems to manage these factors better and could contribute greater flavor intensity to the wine as a significantly higher concentration was found compared to the wine from the other strain. Another polyol is 2,3-butanediol (meso and levo), which can be formed from acetaldehyde and acetoin [44]. So, it makes sense that 2,3-butanediol (meso) is also found in higher proportions in the wine at 9 months with the G1 strain.

Esters are very important compounds in the aroma of wines, and many of them are produced through yeast fermentation [45]. They are the most abundant family of volatile metabolites in qualitative terms [46]. There are three esters that form part of the major volatile compound esters, two of which showed significant differences at 9 months. Higher concentrations of ethyl acetate were found in the wine labeled N62, but these did not exceed 80 mg/L. This may have enhanced the wine’s organoleptic properties [47]. Although the olfactory threshold of 150 mg/L was not exceeded in any wine, ethyl lactate was found to be significantly upregulated in wine with strain N62. This ester may therefore not contribute to the aroma of the wines [48]. The esters that are part of the minor volatile compounds are divided into two groups. Ethyl esters are produced from the byproducts of medium-chain fatty acids [29]. And acetate esters are formed from acetic acid or acetyl CoA and higher alcohols [16]. This last group is considered to have a greater effect on the aroma than ethyl esters [29]. As in the study by Garofalo et al. [49], the ethyl esters that originated through yeast metabolism and contributed most to the aroma were ethyl hexanoate and octanoate and the acetate ester known as isoamyl acetate. The concentration of ethyl esters was found to be significantly higher in wine with N62 at 9 months, while that of the acetate ester was lower. These metabolites have the potential to contribute different aromas, but this depends on the interaction of the aroma compounds in wine [49]. One possible reason for the significantly higher concentration of isoamyl acetate in wine with G1 is that it can be formed from leucine, which decreases significantly at 9 months [50]. Other ethyl esters that were also important in the aroma of wine, as in the study by Voce et al. [51], were ethyl butanoate, ethyl-2-methylbutanoate, and ethyl-3-methylbutanoate, which exceeded the odor threshold at 9 months, except for ethyl-2-methylbutanoate in wine with G1. Ethyl isobutanoate decreases in both wines during fermentation but increases at the end of the process. These results are consistent with Ruiz-Moreno et al. [52], so this ester could be considered a marker of CO_2_ overpressure.

Terpenes can interact with the cell wall of yeast, producing less aroma and flavor [53]. In the wine with the G1 strain, a high concentration of these aromatic compounds was produced halfway through fermentation (3 bar pressure), but no significant differences were found with the concentrations at 9 months. This could be explained by the high metabolic cell wall activity of this strain [5]. In contrast, significant differences were found in the wine with the N62 strain, in which most of these compounds increased at 9 months. As shown in Figure 6b, the concentration of terpenes was higher in this wine than in the wine with G1. This could be because N62 is β-glucosidase-positive (see Materials and Methods section).

Lactones are valued compounds in wines, contributing floral and fruity aromas [54]. Hydroxy acids through intramolecular esterification produce these compounds during alcoholic fermentation [55]. Butyrolactone, which can contribute to a fruity or caramel aroma [56], was the most abundant metabolite in both wines, exceeding the perception threshold and increasing significantly throughout fermentation.

Furans are formed from the breakdown of sugars and can contribute yeasty and toasted aromas to wines [57]. Three furans were identified, and in two of them no significant differences were found at 9 months between the two wines, although 2-acetylfuran was only found at 9 months. Pentylfuran was only found in the BW and in the wines with strain N62. When the wine reached 3 bars of pressure, it increased significantly and then remained at that concentration.

Yeasts use nitrogen compounds for growth and nitrogen balance [58]. The concentration of most of these compounds increased significantly after 9 months, which could indicate the onset of yeast autolysis [26], as also shown by the low viability data in Table 2. According to Gobert et al. [59], a higher concentration of amino acids leads to a higher concentration of higher alcohols. This relationship was also observed in this study. The amino acids glutamic acid, aspartic acid, and alanine have the ability to enhance flavor [60]. These were found in significantly higher proportions in the wine with N62 at 9 months, so it is possible that the flavor in this wine is enhanced. As can be seen in Figure 6c, most nitrogen-containing compounds are upregulated for wine with N62, so it could be said that this strain was more active in the biosynthetic processes of amino acids [61]. Specifically, biogenic amines originate from the amination and transamination of aldehydes and ketones or from the decarboxylation of the corresponding amino acids [62]. It appears that these processes were not carried out on a large scale because high values of these nitrogen compounds were not obtained.

## 4. Materials and Methods

### 4.1. Yeast Strains and Characterization

Strains G1 and N62 are *S. cerevisiae*, isolated from veils of flor of fine wine subjected to biological ageing in the Pérez Barquero winery Protected Designation of Origin (P.D.O.) from Montilla-Moriles, Córdoba, Spain. Both strains were evaluated for ethanol tolerance, CO_2_ production, turbidity, and flocculation using synthetic media (3 g/L yeast extract (Oxoid, Basingstoke, Union Kingdom), 5 g/L peptone (Oxoid), 10 g/L dextrose (Oxoid), and appropriated wine ethanol concentration (0–14%, *v*/*v*). Both strains were inoculated separately in the medium described above in plastic tubes. If they grew at certain ethanol concentrations in the form of turbidity or flocs, they were considered to tolerate that ethanol concentration. Both strains produced flocs at high ethanol concentrations. If, after several days, the plastic cap on the tubes containing the medium and the strains came off, it was considered that they were producing CO_2_. This occurred for both strains. In addition, on activity β-glucosidase detection medium (5 g/L arbutin (Sigma-Aldrich Chemie GmbH, Taufkirchen, Germany), 1 g/L yeast extract (Oxoid), 20 g/L agar (PanReac, Castellar del Vallès, Barcelona, Spain) and 0.2% of a 1% (*w*/*v*) ferric chloride solution (Oxoid) were used. The medium described above was dispensed into Petri dishes and inoculated with 100 µL of liquid YPD medium (1% yeast extract, 2% peptone, and 2% glucose) from each strain. The dishes were incubated at 28 °C for 15 days. Dark black media were considered positive; only strain N62 was found to be positive.

### 4.2. Acclimatation Process and Sparkling Wine Production

These yeast strains were adapted to a medium containing 4% sucrose (*w*/*v*) (PanReac), 0.72 g/L DAP (diammonium phosphate), 0.5% (*w*/*v*) yeast extract (Oxoid), and wine alcohol. The percentage of wine alcohol in the medium was increased progressively to 12% (*v*/*v*).

The characteristics of the BW were density 0.9994 g/mL; 10.01% *v*/*v* ethanol; glucose and fructose 0.55 g/L; total sulfur dioxide 75 mg; free sulfur dioxide 11 mg; pH 3.08; and total acidity of 5.30 g/L. The BW was provided by the company Pérez Barquero. Sparkling wine was produced according to the traditional method. In each bottle 750 mL of BW was added, with 24 g/L of sucrose, 0.72 DAP, and an inoculum of 1.5 × 10^6^ cells/mL of each yeast strain. The bottles were sealed with a stopper and a crown cap. These were placed in a chamber at a controlled temperature and humidity (12 ± 1 °C and 75% humidity). The second fermentation in bottle was monitored by changes in the endogenous pressure of CO_2_ using an internal aphrometer (Oenotilus, Station Oenotechnique de Champagne, Epernay, France). When the pressure reached 3 bars and 9 months had elapsed, three random bottles were collected for each yeast (three biological replicates) at each point.

### 4.3. Chemical Analysis and Oenological Parameters of Wines

Oenological parameters such as volatile acidity, total acidity, ethanol content (%, *v*/*v*), reducing sugars, and pH were quantified according to OIV methods [63]. Lactic and malic acid levels were determined by reflectometry using Reflectoquant™ (Merck^®^, Darmstadt, Germany).

### 4.4. Cell Counting: Total and Viable

The total number of cells was determined by counting yeast suspension diluted to 10^−2^ using a Thoma chamber under a binocular optical microscope equipped with a 40× objective lens. Cell viability was assessed by plating 100 µL of the diluted suspension onto YPD agar medium in Petri dishes (composed of 1% yeast extract, 2% peptone, 2% dextrose, and 2% agar). After incubating the plates for 48 h at the optimal growth temperature of 28 °C, viable yeast colonies were counted. All measurements were performed in triplicate.

### 4.5. Analysis of Wine Volatiles

The analysis of volatile compounds was carried out following the methodology described by Martínez-García et al. [64]. Gas chromatography was used with an Agilent 6890 GC device (Agilent technologies, Santa Clara, CA, USA) equipped with a Flame Ionization Detector (FID) and a capillary column CP-WAX 57 CB (60 m, 0.25 mm, and 0.4 µm of film thickness) by direct injection of the different wine samples to analyze the major volatile compounds and polyols. A total of 10 mL of sample was analyzed together with 1 mL of a 1.018 g/L of 4-methyl-2-pentanol (CAS 108-11-2) solution as internal standard and 0.2 g of solid calcium trioxide carbonate (CaCO_3_). The solution was then subjected to an ultrasonic bath for 30 s and centrifuged at 5000 rpm for 10 min (2 °C). Then 0.7 µL of the supernatant was injected into the gas chromatograph inlet. Compounds such as methanol, higher alcohols, acetaldehyde, acetoin, ethyl esters, glycerol, and 2,3-butanediol were quantified using a calibration table constructed with standard solutions of known concentrations.

The minor volatile compounds were quantified and identified using the platform SBSE-TD-GC-MS (Stir Bar Sorptive Extraction–Thermal Desorption–Gas Chromatography–Mass Spectrometry). This platform includes an Agilent-7890A GC coupled to an MSD 5975C (Wilmington, DE, USA) and a Gerstel Multi-Purpose Sampler (MPS) (GmbH & Co. KG—Mülheim an der Rhur, Germany). The software programs used were ChemStation v. 02.02.1431 from Agilent and Maestro from Gerstel, the first as the chromatographic data processor and the second as the platform control. Using the SBSE technique, minor volatile compounds were extracted, for which a twister was used (10 mm long and 0.5 mm thick film) coated with polydimethylsiloxane (PDMS), which preferentially adsorbs low polar and apolar compounds. The next step consisted of placing the rotary shaker in the vial and shaking at 1200 rpm and 20 °C for 120 min to promote adsorption of the compounds in a Variomag Multipoint 15 magnetic stirrer (Thermo Fisher Scientific, Waltham, MA, USA). Then, the twister was removed, and water was then rinsed and dried and placed in a desorption tube to be transferred by MPS to the Thermal Desorption Unit (TDU), where the volatiles were desorbed and transferred to the GC system. A HP-5MS-fused silica capillary column (60 m × 0.25 mm i.d. and 0.25 μm film) from Agilent Technologies was used along with an initial oven temperature of 50 °C (2 min), which was then increased at a rate of 4 °C/min to 190 °C for 10 min. The MSD operated at 70 eV in the electron impact (EI) mode, with a mass range of 35–550 Da at a temperature of 150 °C. All samples were analyzed in triplicate. Quantification of minor volatile compounds was performed using a calibration table built with standard solutions, containing a known concentration of each compound.

All volatile compounds (majority and minority) were identified and confirmed by GC-MS using the same Agilent 7890-MSD 5975C (Agilent technologies, Santa Clara, CA, USA, and Wilmington, DE, USA) described before and the same capillary column and settings for temperature and carrier helium gas programs used for their analysis. Compound identification was performed by comparing the peak data of compounds with mass spectra libraries NIST08 and Wiley7 and consulting the NIST database from the Web of Chemistry. Another identification was performed by subjecting a mixture of commercially available pure compounds to the same analytical conditions as the samples. Reagents and pure chemical compounds for identification and quantification were provided by Sigma-Aldrich (St. Louis, MO, USA) and Merck (Darmstadt, Germany).

### 4.6. Quantification of Nitrogen Compounds

Quantitative analysis of amino acids (14 detected), biogenic amines (3 detected), and ammonium chloride was performed via high-performance liquid chromatography using an adapted derivatization methodology with diethyl ethoxymethylenemalonate (DEEMM) (Sigma-Aldrich; St. Louis, MO, USA), following the protocol described by Gómez-Alonso et al. [65]. Samples were collected in triplicate in 1.5 mL Eppendorf tubes and stored at −20 °C prior to analysis. To achieve derivatization of the compounds, 0.250 mL of untreated sample, 0.250 mL of Milli-Q water, 0.500 mL of methanol, 0.750 mL of 1 M borate buffer (pH = 9), 0.050 mL of L-2-aminoadipic acid (1 g/L) as internal standard, and 0.003 mL of DEEMM were combined in a tube. Then, the dissolution was subjected to ultrasonication for 30 min followed by heating at 70 °C for 2 h. The quantitative analysis was carried out using an Agilent HPLC 1260 Infinity system (Palo Alto; Santa Clara, CA, USA) equipped with an ACE C18-HL column (250 mm × 4.6 mm, 5 μm particle size) thermostated at 16 °C, employing a binary gradient of mobile phases A and B, as detailed by Gómez-Alonso et al. [65]. The detection of analytes was achieved using a photodiode array detector set at a wavelength of 280 nm.

It is acknowledged that the methodology and scope of the method used in this study for the determination and quantification of nitrogen compounds by HPLC was rigorously developed and validated for the simultaneous determination and quantification of 23 amino acids, 7 biogenic amines, and ammonium ions.

It is important to note that, of all the compounds that were validated by the method, not all were present in the samples above the detection threshold of the method. Appendix A contains only the nitrogen compounds (amino acids, biogenic amines, and ammonium) that were present and quantifiable in the samples.

### 4.7. Statistical Analysis of Data

All metabolites were subjected to a two-way analysis of variance (ANOVA), the least significant test (Fisher’s LSD test), and correction FDR test to establish significant differences between them. Differences at *p* < 0.05 were considered to be statistically significant. The data matrices of the concentrations of major volatile compounds and polyols, those of minor volatile compounds, and those of nitrogen compounds were statistically autoscaled. MetaboAnalyst was used to create the Principal Component Analysis (PCA), heat maps and variables of importance in the projection (VIP). Statistical analysis was used for the Prism 9.0 software’s (GraphPad Software, La Jolla, CA, USA) kinetics of second fermentation, volcano plot, ANOVA, and Fisher’s LSD test. One-way ANOVA and Tukey’s HSD test (q < 0.05) were used for the volcano plot.

## 5. Conclusions

This study demonstrates that both flor yeast strains, G1 and N62 of *Saccharomyces cerevisiae* are viable alternatives to the traditional Méthode Champenoise second fermentation, each providing distinct metabolic traits that enhance product diversification in sparkling wine production. Strain N62 showed faster fermentation kinetics—reaching target CO_2_ pressure in approximately half the time required by strain G1—while both achieved comparable final pressures after nine months of ageing. Comprehensive metabolomic analysis confirmed that strain G1 is characterized by a greater ability to produce aromatic higher alcohols and esters such as 3 methyl 1 butanol, 2 methyl 1 butanol, acetaldehyde, and isoamyl acetate, which may contribute fruity and floral complexity. In contrast, strain N62 displayed enhanced glycerol synthesis, higher yields of amino acids (notably glutamic, aspartic, and alanine), and marked β glucosidase activity, resulting in elevated terpene and ethyl ester levels that may enrich mouthfeel and fruity intensity.

During nine months of ageing, both strains showed significant increases in nitrogen compounds, indicating successful autolysis and release of macromolecules that may contribute to aromatic and structural complexity. The distinct metabolic fingerprints of these flor yeasts underline their adaptability to the pressurized conditions of sparkling wine fermentation and their potential to express regional sensory typicity.

Overall, these findings confirm the oenological potential of flor yeasts—traditionally associated with biologically aged wines—as innovative biotechnological tools for sparkling wine production. Their physiological robustness, metabolomic diversity, and fermentation efficiency under pressure conditions expand the currently limited spectrum of available tirage yeasts. Practically, strain G1 may be preferred for wines seeking aromatic complexity, whereas strain N62 is more suited for styles emphasizing volume, texture, and freshness. This work contributes to the sustainable diversification of sparkling wine yeast resources and supports the broader objective of increasing microbial biodiversity in premium winemaking.

## Figures and Tables

**Figure 1 ijms-26-10457-f001:**
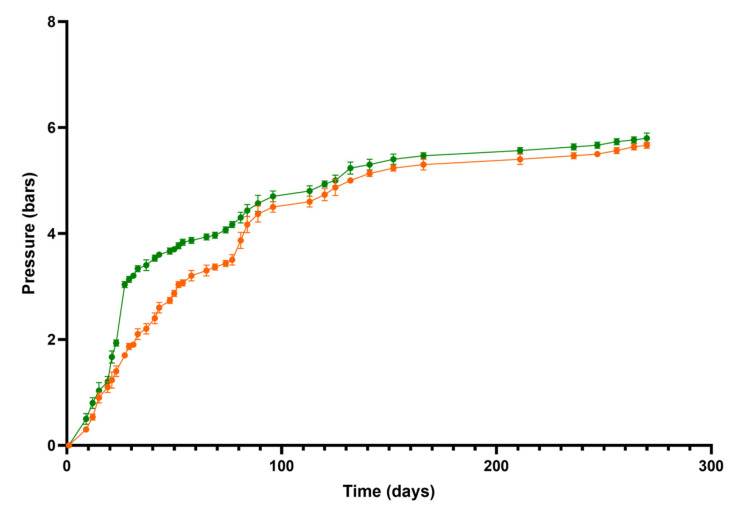
Monitoring of the second fermentation produced by CO_2_ pressure carried out by yeast strains: wine fermented with strain G1 (orange) and wine fermented with strain N62 (green).

**Figure 2 ijms-26-10457-f002:**
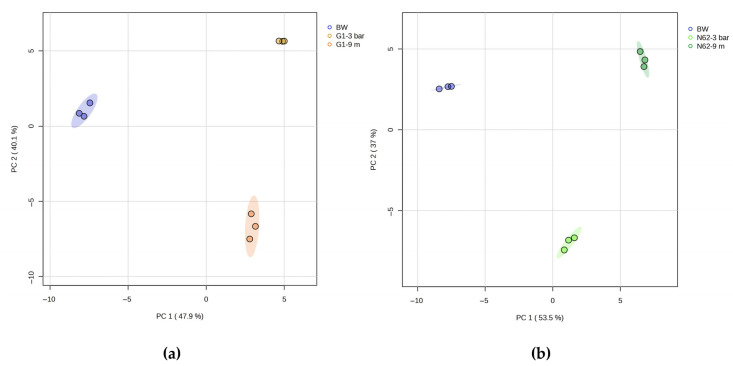
Principal component analysis (PCA) carried out from the data matrix of the metabolites quantified (**a**) base wine (BW, blue), wine with G1 strain at 3 bars of pressure (G1-3 bar, ocher), and wine with strain G1 at 9 months (G1-9 m, orange) (**b**) base wine (BW, blue), wine with N62 strain at 3 bars of pressure (N62-3 bar, light green), and wine with strain N62 at 9 months (N62-9 m, green).

**Figure 3 ijms-26-10457-f003:**
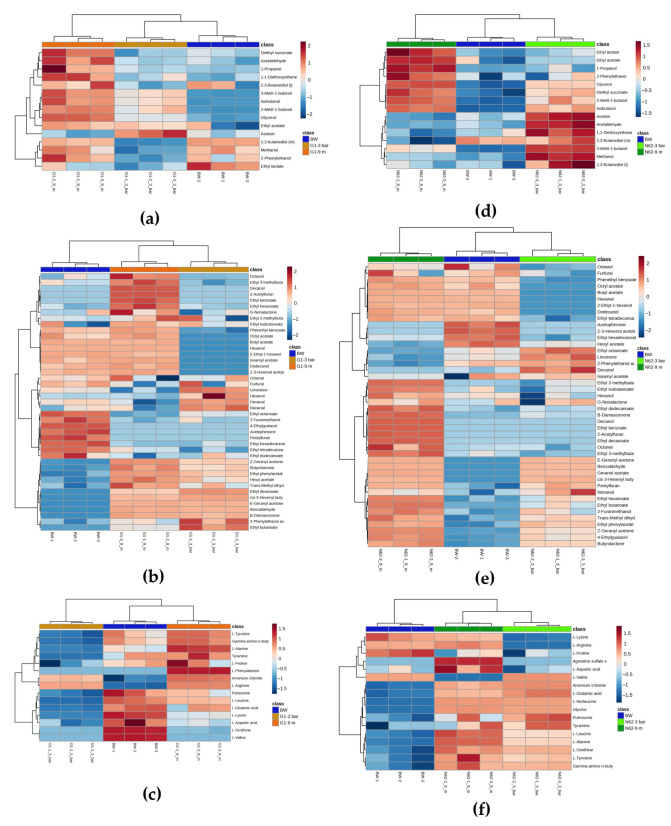
Visualization of the heat map of the quantified and normalized data of the different wines. Base wine (BW, blue), wine inoculated with G1 at 3 bar pressure (G1-3 bar, ocher), wine inoculated with G1 at 9 months (G1-9 m, orange), wine inoculated with N62 at 3 bar pressure (N62-3 bar, light green), and wine inoculated with N62 at 9 months (N62-9 m, green). (**a**–**c**) Major volatile compounds and polyols, minor volatile compounds, and nitrogen compounds of wine with G1, respectively. (**d**–**f**) Major volatile compounds, minor volatile compounds, and nitrogen compounds of wine with N62, respectively.

**Figure 4 ijms-26-10457-f004:**
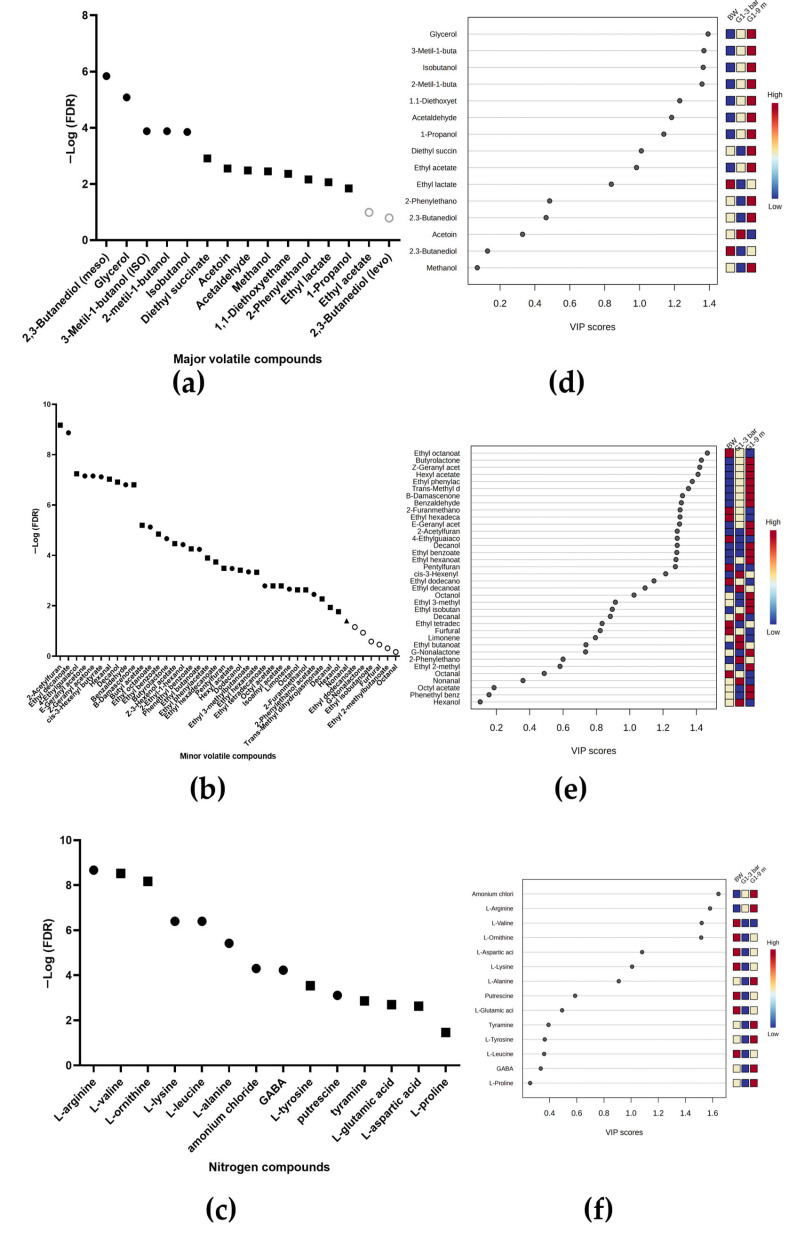
Analysis of ANOVA and Fisher’s test (**a**–**c**) comparing metabolites of base wine (BW), wine inoculated with G1 at 3 bar pressure, and wine inoculated with G1 at 9 months. Circles indicate significant differences among three wines. Squares indicate no significant differences between two wines, but they do show differences between the third wine. Triangles indicate differences between only two wines. Gray circles represent no significant differences. (**d**–**f**) Variable importance in projection (VIP) analysis for the same three wine groups.

**Figure 5 ijms-26-10457-f005:**
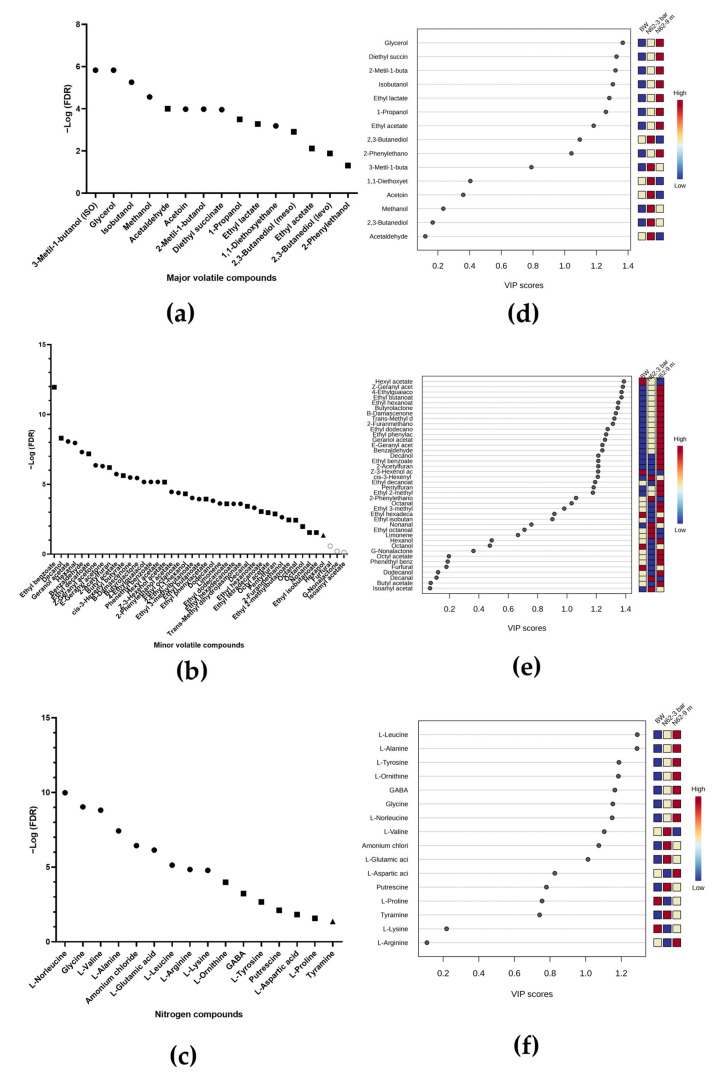
Analysis of ANOVA and Fisher’s test (**a**–**c**) between metabolites of base wine (BW), wine inoculated with N62 at 3 bar pressure, and wine inoculated with N62 at 9 months. The circles indicate that the metabolite shows significant differences between the three wines. The squares indicate that the metabolite shows no significant differences between two wines, but they do show differences between the third wine. The triangles indicate that the metabolite shows significant differences between only two wines. The gray circles indicate that the metabolite shows no significant differences between any wines. (**d**–**f**) The analysis of the variables of importance in the projection between metabolites of BW, wine inoculated with N62 at 3 bar pressure, and wine inoculated with N62 at 9 months.

**Figure 6 ijms-26-10457-f006:**
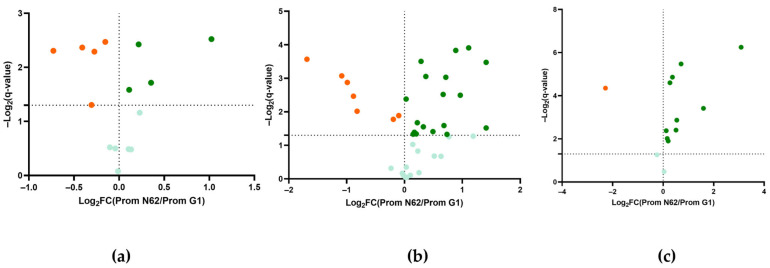
Volcano plot of metabolites at nine months with significant differences at q-value < 0.05. Upregulated metabolites with strain N62 correspond to the green spheres (LFC > 0), and the upregulated metabolites with strain G1 correspond to the orange spheres (LFC < 0). Metabolites without significant differences are shown in light green. (**a**) Comparison of major volatile compounds and polyols. (**b**) Comparison of minor volatile compounds. (**c**) Comparison of nitrogen compounds.

**Table 1 ijms-26-10457-t001:** General chemical variables analyzed in wines when they reached 3 bars of pressure and after 9 months.

	G1 (3 Bars)	G1 (9 Months)	N62 (3 Bars)	N62 (9 Months)
pH	3.022 ^a^ ± 0.071	3.276 ^b^ ± 0.061	3.132 ^a^ ± 0.061	3.302 ^b^ ± 0.082
Titratable acidity (g/L)	4.343 ± 0.613	4.022 ± 0.533	4.260 ± 0.111	3.949 ± 0.105
Ethanol (% *v*/*v*)	11.929 ^a^ ± 0.080	12.134 ^b^ ± 0.090	11.100 ^c^ ± 0.080	11.987 ^ab^ ± 0.090
Volatile acidity (g/L)	0.224 ^a^ ± 0.021	0.282 ^a^ ± 0.023	0.421 ^b^ ± 0.111	0.497 ^b^ ± 0.131
Reducing sugar (g/L)	0.235 ^a^ ± 0.004	0.197 ^b^ ± 0.004	0.221 ^c^ ± 0.003	0.178 ^d^ ± 0.003
Malic acid (g/L)	0.220 ^a^ ± 0.010	0.199 ^b^ ± 0.010	0.010 ^c^ ± 0.000	0.010 ^c^ ± 0.000
Lactic acid (g/L)	1.213 ^a^ ± 0.010	1.989 ^b^ ± 0.111	0.330 ^c^ ± 0.010	1.334 ^d^ ± 0.032

^a,b,c,d^ Different letters in the same row indicate statistical differences of the normalized and scaled data at 0.05 level according to Fisher’s least significant difference test, represented in the table as homogeneous groups (HGs).

**Table 2 ijms-26-10457-t002:** Viable and total cell count of the two strains at 3 bars of pressure and after 9 months.

	G1 (3 Bars)	G1 (9 Months)	N62 (3 Bars)	N62 (9 Months)
Viable (× 10^6^ cells/mL)	0.730 ^a^ ± 0.035	0.060 ^b^ ± 0.010	3.900 ^c^ ± 0.265	0.830 ^a^ ± 0.040
Total (× 10^6^ cells/mL)	21.000 ^a^ ± 1.730	23.000 ^a^ ± 1.832	46.300 ^b^ ± 3.210	44.86 ^c^ ± 2.340

^a,b,c^ Different letters in the same row indicate statistical differences of the normalized and scaled data at 0.05 level according to Fisher’s least significant difference test, represented in the table as HGs.

## Data Availability

The original contributions presented in this study are included in the article/Appendix A. Further inquiries can be directed to the corresponding author.

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
