# Peer review of "Metabolic Characterization of Two Flor Yeasts During Second Fermentation in the Bottle for Sparkling Wine Production"

_ijms, 2025, doi:10.3390/ijms262110457_

Round 1
Reviewer 1 Report
Comments and Suggestions for Authors
General comments
This manuscript investigates the metabolic characterization of two Saccharomyces cerevisiae flor yeast strains (G1 and N62) during the second fermentation in bottle for sparkling wine production. The topic is relevant and aligned with the scope of IJMS, particularly for its molecular and metabolomic approach to yeast strain performance. The work is well organized, well executed, the data are extensive, and the conclusions are consistent with the results. However, some aspects will be clarified and corrected before the paper can be accepted.
Major Comments
- Novelty and Scientific Context. The research group has published related studies on these flor yeasts for sparkling wines. Therefore, the originality relative to previous works by the same group should be explicitly clarified. Please emphasize what is the novelty of the manuscript in the Introduction section.
- Materials and Methods. The manuscript does not provide any information about the methods used to determine total cell count and viability. Since these methods are important to the results, they should be included in the text.
- Section 2.1. Tables 1 and 2 present data on various oenological parameters and cell viability at two key points in the process. Since three biological replicates were performed, a statistical analysis of these variables would be beneficial for the manuscript.
- Figures 3-5. They are visually complex and difficult to read. Consider providing higher-resolution figures. They are visually complex and difficult to read. Consider providing higher-resolution figures. I don't understand Figures 4a-c and 5a-c. What is represented on the X-axis? Does each analyzed compound correspond to a number? That it's right, the compound identification should be included in Tables S1 and S2.
- Figure 6. Check for consistency between the text and the figure. For example, Figure 6b shows seven compounds that are up-regulated in G1 wines, but the authors indicate in the text that there are six. To improve the interpretation, the footnote for this figure should be changed as follows: "Volcano plot of metabolites at nine months with significant differences at a p-value of less than 0.05. Up-regulated metabolites with strain N62 correspond to green spheres (LFC > 0), and up-regulated metabolites with strain G1 correspond to orange spheres (LFC < 0). Metabolites without significant differences are shown in light green. (a) Comparison of major volatile compounds and polyols. (b) Comparison of minor volatile compounds. (c) Comparison of nitrogen compounds."
- Tables S1 and S2. To better understand the work carried out, please include the titles of the tables and the units. What do the abbreviations BW-1, BW-2, and BW-3 refer to?
- Conclusions Section. The Conclusions are too extensive and read like a summary of the Discussion. Please condense this section into 2-3 paragraphs emphasizing the main findings and their practical relevance.
- Graphical Abstract. Consider including a simple schematic summarizing the experimental design.
Author Response
File attached

Reviewer 2 Report
Comments and Suggestions for Authors
General Comments
The manuscript presents an interesting study on the use of flor Saccharomyces cerevisiae strains for the second fermentation of sparkling wines. The topic is relevant for microbial diversity and innovation in oenology. However, several aspects of the manuscript require clarification, restructuring, and more rigorous data treatment before it can be considered for publication.
Specific Comments
Abstract
- The statement on the “growing global demand for sparkling wines” should be supported by quantitative data (e.g., OIV statistics).
Given that global alcoholic beverage consumption is decreasing and there is a strong trend toward low- or non-alcoholic alternatives, this claim should be precisely substantiated. - Include at least one or two numerical data points in the abstract to better capture the reader’s attention and provide context.
Introduction
- The same quantitative information on global demand should also be included here to strengthen the background.
- The authors position the study as innovative because flor yeasts are used for secondary fermentation in sparkling wine. However, several studies (e.g., Martínez-García et al., 2020; García-García et al., 2025), both cited by the authors, have already evaluated flor yeasts in this context. Please clearly state the novelty of the current study compared to these previous works.
- The introduction contains some repetition of “sparkling wine” trends and general market information. The section could be restructured for better logical flow.
- Please ensure that the structure of the introduction aligns with the journal’s author guidelines, separating the rationale, objectives, and hypothesis more clearly.
Materials and Methods
- Section 4.1 lacks sufficient detail on yeast characterization. The authors list the growth media but do not explain how the yeasts were evaluated (e.g., growth rate, flocculation index, optical density, visual biofilm formation, etc.). Please provide a more detailed description of the characterization procedures and criteria.
- The manuscript states that PCA, ANOVA, and Fisher’s LSD tests were used, but no multiple testing correction(such as FDR or Bonferroni) is mentioned. Given the large number of metabolites, the potential for false positives (Type I errors) is significant. Please describe or apply an appropriate correction method.
Results and Discussion
- The discussion occasionally over-interprets metabolic correlations without direct biochemical or enzymatic evidence. For example, the interpretation of amino acid conversion pathways should be presented more cautiously unless supported by enzyme activity or transcriptomic data.
- In line 306, the manuscript claims that an amino acid is quantitatively higher at month 9, but it is unclear if all amino acids were actually quantified. If Section 4.5 indeed includes quantification of all amino acids, please extend the results and provide more detailed interpretation accordingly.
- Sensory attributes such as “floral,” “toasted,” and “fruity” are described, but no sensory analysis has been conducted. Please rephrase or remove sensory claims that are not directly supported by data.
Figure 3 is difficult to read: the labels and text are blurry and nearly illegible in printed form (and barely readable in electronic form). Please improve figure resolution and increase label size or separate subpanels for clarity.
Author Response
File attached

Round 2
Reviewer 1 Report
Comments and Suggestions for Authors
Thank you to the authors for their efforts. According to the reviewer's suggestions, the manuscript has been significantly improved in terms of clarity, methodological transparency, and visual presentation.
Reviewer 2 Report
Comments and Suggestions for Authors
The figures seem at some points still blurry but these points will be corrected in the production.
Except of that all critical points has been revised, thank you very much.